# GENERATIVE LATENT FLOW

## ABSTRACT

In this work we propose Generative Latent Flow (GLF), an algorithm for generative modeling of data distributions. GLF uses an auto-encoder to learn latent representations of the data, and a normalizing flow to map the distribution of the latent variables to that of a standard Gaussian. GLF can be seen as a variational auto-encoder, with normalizing flow prior, and a vanishing limit of the pixel-wise variance of the data. We carefully study this relationship and the pros and cons of using an auto-encoder vs. a variational auto-encoder. In contrast to a number of other auto-encoder based generative models, which use various regularizers to encourage the encoded latent distribution to match the prior distribution, our model explicitly constructs a mapping between these two distributions, leading to better density matching while avoiding over regularizing the latent variables. We compare our model with several related techniques, and show that under standard quantitative evaluations, our method achieves state-of-the-art sample quality and diversity among AE based models on commonly used datasets and is competitive with GANs' benchmarks.

## 1 INTRODUCTION

Generative models have attracted much attention in the literature on deep learning. These models are used to formulate the distribution of complex data as a function of random noise passed through a network, so that rendering samples from the distribution is particularly easy. The most dominant generative models are Generative Adversarial Networks (GANs) (Goodfellow et al., 2014), as they have exhibited impressive performance in generating high quality images (Radford et al., 2015; Brock et al., 2018) and in other vision tasks (Zhu et al., 2017; Ledig et al., 2017). Despite their success, training GANs can be challenging, partly because they are trained by solving a saddle point optimization problem formulated as an adversarial game. It is well known that training GANs is unstable and sensitive to hyper-parameter settings (Salimans et al., 2016; Arora et al., 2017), and sometimes training leads to mode collapse (Goodfellow, 2016). Although there have been multiple efforts to overcome the difficulties in training GANs (Arjovsky et al., 2017; Metz Luke & Sohl-Dickstein, 2017; Srivastava et al., 2017; Miyato et al., 2018), researchers are also actively studying non-adversarial methods that are known to be less affected by these issues.

Some models explicitly define $p(x)$, the distribution of the data, and training is guided by maximizing the data likelihood. One approach is to express the data distribution in an auto-regressive pattern (Papamakarios et al., 2017; Oord et al., 2016); another is to express it as an invertible transformation of a simple distribution using the change of variable formula, where the invertible transformation is defined using a normalizing flow network (Dinh et al., 2014; 2016; Kingma & Dhariwal, 2018). While being mathematically clear and well defined, normalizing flows keep the dimensionality of the original data in order to maintain bijectivity. Consequently, they cannot provide low-dimensional representations of the data and training is computationally expensive. Considering the prohibitively long training time and advanced hardware requirements in training large scale flow models such as (Kingma & Dhariwal, 2018), we believe that it is worth exploring the application of flows in the low dimensional representation spaces rather than for the original data.

Another class of generative models employs an encoder-decoder structure and low dimensional latent variables to represent and generate the data. An encoder is used to produce estimates of the latent variables corresponding to a particular data point, and samples from a predefined prior distribution on the latent space are passed through a decoder to produce new samples from the data distribution. We call these auto-encoder (AE) based models, of which variational auto-encoders (VAEs) are perhaps

the most influential (Kingma & Welling, 2013; Rezende et al., 2014). VAEs use the encoder to produce approximations to the posterior distribution of the latent variable given the data, and the training objective is to maximize a variational lower bound of the data log likelihood. VAEs are easy to train, but their generation quality still lies far below that of GANs, as they tend to generate blurry images (Dosovitskiy & Brox, 2016).

Whereas the original VAE uses a standard Gaussian prior, it can be extended by introducing a learnable parameterized prior distribution. There have been a number of studies in this direction (see section 2), some of which use a normalizing flow parameterization, where the prior is modeled as a trainable continuous bijective transformation of the standard Gaussian. We carefully study this method, and make the surprising novel observation that in order to produce high quality samples, it is necessary to significantly increase the weight on the reconstruction loss. This corresponds to decreasing the variance of the observational noise of the generative model at each pixel, where we are assuming the data distribution is factorial Gaussian conditioned on the output of the decoder, which yields the MSE as the reconstruction loss. It is important to note that increasing this weight alone without access to a trainable prior does not consistently improve generation quality.

We show that as this weight increases, we approach a vanishing noise limit that corresponds to a deterministic auto-encoder. This leads to a new algorithm we call Generative Latent Flow (GLF), which combines a deterministic auto-encoder that learns a mapping to and from a latent space, and a normalizing flow that matches the standard Gaussian to the distribution of latent variables of the training data produced by the encoder.

Our contributions are summarized as follows: i) we carefully study the effects of equipping VAEs with a normalizing flow prior on image generation quality as the weight of the reconstruction loss is increased. ii) Based on this finding, introduce Generative Latent Flow, which uses auto-encoders instead of VAEs. iii) Through standard evaluations, we show that our proposed model achieves state-of-the-art sample quality among competing AE based models, and has the additional advantage of faster convergence.

## 2 RELATED WORK

In general, in order for an AE based model with encoder-decoder structure to generate samples resembling the training data distribution, two criteria need to be ensured: (a) the decoder is able to produce a good reconstruction of a training image given its encoded latent variable $z$; and (b) the empirical latent distribution $q(z)$ of $z$'s returned by the encoder is close to the prior $p(z)$. In VAEs, the empirical latent distribution is often called aggregated or marginal posterior: $q(z) = \mathbb{E}_{x \sim p_{data}} [q(z|x)]$. While (a) is mainly driven by the reconstruction loss, satisfying criterion (b) is more complicated. Intuitively, criterion (b) can possibly be achieved by designing mechanisms that either modify the empirical latent distribution $q(z)$, or conversely modify the prior $p(z)$. There is plenty of previous work in both directions.

**Modifying the empirical latent distribution $q(z)$:** In the classic VAE model, $D_{\mathrm{KL}}(q(z|x)\|p(z))$ in the ELBO loss can be decomposed as $D_{\mathrm{KL}}(q(z)\|p(z))$ plus a mutual information term as shown in (Hoffman & Johnson, 2016). Therefore, VAEs modify $q(z)$ indirectly through regularizing the posterior distribution $q(z|x)$. Several modifications to VAE's loss (Chen et al., 2018; Kim & Mnih, 2018), which are designed for the task of unsupervised disentanglement, put a stronger penalty specifically on the mismatch between $q(z)$ and $p(z)$. There are also attempts to incorporate normalizing flows into the encoder to provide more flexible approximate posteriors (Rezende & Mohamed, 2015; Kingma et al., 2016; Berg et al., 2018). However, empirical evaluation shows that VAEs with flow posteriors do not reduce the mismatch between $q(z)$ and $p(z)$ (Rosca et al., 2018). Furthermore, as of yet, all these modifications to VAEs have not been shown to improve generation quality. Adversarial auto-encoders (AAEs) (Makhzani et al., 2015) and Wasserstein auto-encoders (WAEs) (Tolstikhin et al., 2017) use an adversarial regularizer or MMD regularizer (Gretton et al., 2012) to force $q(z)$ to be close to $p(z)$. WAEs are shown to improve generation quality, as they generate sharper images than VAEs do.

**Modifying the prior distribution $p(z)$:** An alternative to modifying the approximate posterior is using a trainable prior. (Tomczak & Welling, 2017; Klushyn et al., 2019; Bauer & Mnih, 2018) propose different ways to approximate $q(z)$ using a sampled mixture of posteriors during training,

and then use the approximated $q(z)$ as the prior in the VAE. This is a natural way to let the prior match $q(z)$, however, these methods have not been shown to improve generation quality. Two-stage VAE (Dai & Wipf, 2019) introduces another VAE on the latent space defined by the first VAE to learn the distribution of its latent variables. VQ-VAE (Oord et al., 2017) first trains an auto-encoder with discrete latent variables, and then fits an auto-regressive prior on the latent space. GLANN (Hoshen et al., 2019) learns a latent representation by GLO (Bojanowski et al., 2017) and matches the densities of the latent variables with an implicit maximum likelihood estimator (Li & Malik, 2018). RAE+GMM (Ghosh et al., 2019) trains a regularized auto-encoder (Alain & Bengio, 2014) and fits a mixture of Gaussian distribution on the latent space. Note that all these methods involve two stage-training, which means that the prior distribution is fitted after training the variational or deterministic auto-encoder. They have been shown to improve the quality of the generated images. VAEs with a normalizing flow as a learnable prior (Chen et al., 2016b; Huang et al., 2017) also fall into this category. Since these are the main focus of this paper, we discuss them in detail in Section 3.2.

We note that modifications of VAEs with a normalizing flow posterior have been extensively studied. In contrast, VAEs with flow prior have attracted much less attention. (Huang et al., 2017) briefly discusses this model to solve the distribution mismatch in the latent space, and recently (Xu et al., 2019) shows the advantages of learning a flow prior over learning a flow posterior. However, these papers only focus on improvements of the data likelihood. Here we study the model from the perspective of the effects of the normalizing flow prior on sample generation quality, leading to some important and novel observations.

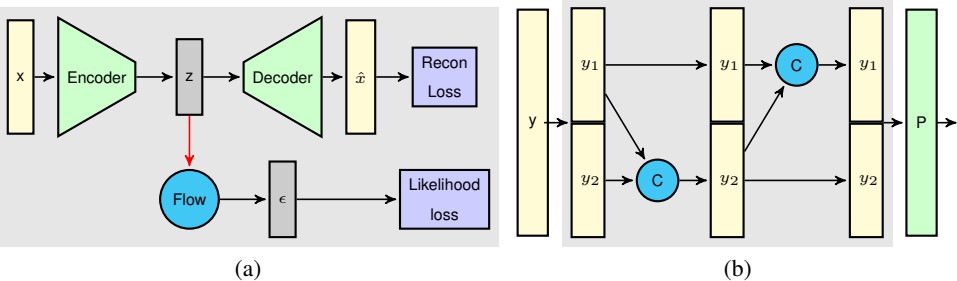

(a)                                                                          (b)

Figure 1: (a) Illustration of the GLF model. The red arrow contains a stop gradient operation (see section 3.3). (b) Structure of one flow block. The input is split into two parts $y = (y_1, y_2)$, go through two coupling layers $C$ (see section 3.1). Finally a random permutation $P$ is applied.

# 3 COMBINING NORMALIZING FLOW WITH AE BASED MODELS

In this section, we discuss the combination of normalizing flow priors with AE based models in detail. We first review normalizing flows in section 3.1, then in section 3.2 we introduce VAEs with normalizing flow prior and present some novel observations with respect to this model. Finally in section 3.3 we propose Generative Latent Flow (GLF) to further simplify the model and improve performance.

## 3.1 REVIEW: NORMALIZING FLOWS

Normalizing flows are carefully-designed invertible networks that map the training data to a simple distribution. Let $z \in \mathcal{Z}$ be an observation from an unknown target distribution $z \sim p(z)$ and $p_\epsilon$ be the unit Gaussian prior distribution on $\mathcal{E}$. Given a bijection $f_\theta : \mathcal{Z} \to \mathcal{E}$, we define a probability model $p_\theta(z)$ with parameters $\theta$ on $\mathcal{Z}$. The negative log likelihood (NLL) of $z$ is computed by the change of variable formula:

$$\mathcal{L}_{\mathrm{NLL}}(f_\theta(z)) \triangleq -\log(p_\theta(z)) = -\left( \log p_\epsilon(f_\theta(z)) + \log \left| \det \left( \frac{\partial f_\theta(z)}{\partial z} \right) \right| \right), \qquad (1)$$

where $\frac{\partial f_\theta(z)}{\partial z}$ is the Jacobian matrix of $f_\theta$. In order to learn the flow $f_\theta$, the NLL objective of $z$ is minimized, which is equivalent to maximizing the likelihood of $z$. Since the mapping is a bijection, sampling from the trained model $p_\theta(z)$ is trivial: simply sample $\epsilon \sim p_\epsilon$ and compute $z = f_\theta^{-1}(\epsilon)$.

The key to designing a normalizing flow model is defining the transformation $f_\theta$ so that the inverse transformation and the determinant of the Jacobian matrix can be efficiently computed. Based on (Dinh et al., 2016), we adopt the following layers to form the flows used in our model.

**Affine coupling layer:** Given $D$ dimensional input data $z$ and $d < D$, we partition the input into two vectors $z_1 = z_{1:d}$ and $z_2 = z_{d+1:D}$. The output of one affine coupling layer is given by $y_1 = z_1$, $y_2 = z_2 \odot \exp(s(z_1)) + t(z_1)$ where $s$ and $t$ are functions from $\mathbb{R}^d \to \mathbb{R}^{D-d}$ and $\odot$ is the element-wise product. The inverse of the transformation is explicitly given by $z_1 = y_1$, $z_2 = (y_2 - t(y_1)) \odot \exp(-s(y_1))$. The determinant of the Jacobian matrix of this transformation is $\det \frac{\partial \mathbf{y}}{\partial \mathbf{z}} = \prod_{j=1}^{d}(\exp[s(z_1)_j])$. Since computing both the inverse and the Jacobian of an affine coupling layer does not require computing the inverse and Jacobian of $s$ and $t$, both functions can be arbitrarily complex.

**Combining coupling layers with random permutation:** Affine coupling layers leave some components of the input data unchanged. In order to transform all the components, two coupling layers are combined in an alternating pattern to form a coupling block, so that the unchanged components in the first layer can be transformed in the second layer. In particular, we add a fixed random permutation of the coordinates of the input data at the end of each coupling block. See Figure 1b for an illustration of a coupling block used in our model.

## 3.2 VAEs with Normalizing Flow Prior

We begin by introducing the training loss of the model. Consider the ELBO loss of standard VAEs with Gaussian prior and posterior ($\eta, \phi$ denote the parameters of encoder and decoder, respectively):

$$\text{ELBO}(\eta, \phi) = \mathbb{E}_{p_{data}(\mathbf{x})} \mathbb{E}_{q_\eta(\mathbf{z}|\mathbf{x})} \left[ \beta \cdot \log p_\phi(\mathbf{x}|\mathbf{z}) + \log p(\mathbf{z}) - \log q_\eta(\mathbf{z}|\mathbf{x}) \right]. \tag{2}$$

The first term is related to the reconstruction loss, while the last two terms can be combined as $D_{\text{KL}}(q(z|x)\|p(z))$. $\beta > 0$ is a hyper-parameter that controls the relative weight of the reconstruction loss and the KL divergence loss. In the standard formulation of VAEs, $p_\phi$ assumes an independent Gaussian distribution with variance 1 at each pixel. The parameter $\beta$ allows us to adjust this variance as $1/\beta$.

If we introduce a normalizing flow $f_\theta$ for the prior distribution, then the prior $p_\theta$ becomes $p_\theta(\mathbf{z}) = p_\epsilon(f_\theta(\mathbf{z})) \left| \det\left( \frac{\partial f_\theta(\mathbf{z})}{\partial \mathbf{z}} \right) \right|$, where $p_\epsilon$ is the standard Gaussian density. Substituting this prior into equation 2, we obtain $\text{ELBO}(\eta, \phi, \theta)$ for VAEs with flow prior:

$$\mathbb{E}_{p_{data}(\mathbf{x})} \mathbb{E}_{q_\eta(\mathbf{z}|\mathbf{x})} \left[ \beta \cdot \log p_\phi(\mathbf{x}|\mathbf{z}) + \log p_\epsilon(f_\theta(\mathbf{z})) + \log \left| \det\left( \frac{\partial f_\theta(\mathbf{z})}{\partial \mathbf{z}} \right) \right| - \log q_\eta(\mathbf{z}|\mathbf{x}) \right]. \tag{3}$$

The second and third terms together are the log-likelihood of $z$ under the prior distribution modeled by the flow (corresponding to the negative of equation 1). The last term corresponds to the entropy of the posterior distribution returned by the encoder. Both the VAE and the normalizing flow are trained by minimizing $-\text{ELBO}(\eta, \phi, \theta)$.

Previous work on VAEs with a flow prior did not consider tuning $\beta$ (which means the reconstruction loss and the KL loss are weighted equally) as they focused on comparing the obtained log likelihoods with those from plain VAEs. However, we observe that when $\beta = 1$, VAEs with a flow prior do not significantly improve the generation quality (see section 4.2 and Table 1). The reason might be that although $p(z)$ is matched with $q(z)$ due to the flow transformation, the decoder is not good enough to reconstruct sharp images (i.e, criterion (a) is not ensured). In contrast, we find that increasing $\beta$ in the objective produces samples with significantly higher quality (see Figure 3). Intuitively, larger weight on the reconstruction loss forces the decoder to produce sharper reconstructed images, while the normalizing flow prior is flexible enough to match the latent distribution.

To the best of our knowledge, we are the first to observe such a relation between the weight of the reconstruction loss and the generation quality of VAEs with flow prior. As $\beta$ increases, two things occur as demonstrated empirically in Section 4.2.1. First the estimated variances from the encoder

decrease, and second the generation quality consistently improves. In the limit, as the posterior variance goes to zero, we obtain a deterministic encoder, leading to a deterministic auto-encoder and a normalizing flow that is used to match the distribution of the latent variables obtained from the data. This is described in detail in the next section.

### 3.3 GENERATIVE LATENT FLOW (GLF)

In an auto-encoder, $z = E_\eta(x)$ is deterministic so that $q(z|x)$ in equation 3 becomes a delta distribution and the entropy term in equation 3 can be removed. The overall training loss is then

$$\mathcal{L}_\beta^{\text{reg}}(\eta, \phi, \theta) = \frac{1}{N} \sum_{i=1}^{N} \Big( \beta \mathcal{L}_{\text{recon}}\big(\mathbf{x}_i, G_\phi(E_\eta(\mathbf{x}_i))\big) + \mathcal{L}_{\text{NLL}}\big(f_\theta(E_\eta(\mathbf{x}_i))\big) \Big), \tag{4}$$

where $\mathcal{L}_{\text{recon}}\big(\mathbf{x}_i, G_\phi(E_\eta(\mathbf{x}_i))\big)$ corresponds to $-\log p_\phi(\mathbf{x}_i|z_i)$ and $\mathcal{L}_{\text{NLL}}\big(f_\theta(E_\eta(\mathbf{x}_i))\big)$ to $-\log p_\theta(z)$ in the negative of equation 3.

As noted in section 3.2, larger $\beta$'s yield better results, in which case the parameters of the auto-encoder are affected almost exclusively by $\mathcal{L}_{\text{recon}}$, while $\mathcal{L}_{\text{NLL}}$ only affects the parameters $\theta$ of the normalizing flow. Therefore, optimizing (4) with extremely large $\beta$ is approximately equivalent to optimizing

$$\mathcal{L}(\eta, \phi, \theta) = \frac{1}{N} \sum_{i=1}^{N} \Big( \mathcal{L}_{\text{recon}}\big(\mathbf{x}_i, G_\phi(E_\eta(\mathbf{x}_i))\big) + \mathcal{L}_{\text{NLL}}\big(f_\theta(\text{sg}\,[E_\eta(\mathbf{x}_i)])\big) \Big), \tag{5}$$

where $\text{sg}[\cdot]$ is the stop gradient operation. The weight parameter $\beta$ is no longer needed because the two loss terms affect independent sets of parameters. We name the model trained by equation 5 as **Generative Latent Flow** (GLF), to highlight that our model applies normalizing flows on latent variables. See Figure 1a for an illustration of the GLF model. We call the model trained by equation 4, without stopped gradient, **regularized GLF**, since the flow acts as a regularizer on the encoder.

Note that when stopping the gradients, GLF can also be trained in two stages, namely an auto-encoder is trained first, and then the flow is trained to map the distribution of estimated latent variables to the standard Gaussian. Empirically, we find that the two-stage training strategy leads to similar performance, so we only focus on one-stage training as it follows our derivation more naturally.

#### 3.3.1 NECESSITY OF STOPPING THE GRADIENTS

The stop gradient operation is necessary when using deterministic auto-encoders. In VAEs with flow prior, the entropy term, which encourages the posterior to have large variance, prevents the degeneracy of the $z$'s. However, when using a deterministic encoder, if we let gradients of $\mathcal{L}_{\text{NLL}}$ back propagate into the latent variables, training can lead to degenerate $z$'s produced by the encoder $E_\eta$. This is because $f_\theta$ has to transform the $z$'s to unit Gaussian noise, so the smaller the scale of the $z$'s, the larger the magnitude of the log-determinant of the Jacobian. Since there is no constraint on the scale of the output of $E_\eta$, the Jacobian term can dominate the entire objective. While the latent variables cannot become exactly 0 because of the presence of the reconstruction loss, the extremely small scale of $z$ may cause numerical issues that lead to severe fluctuations. In summary, we stop the gradient of $\mathcal{L}_{\text{NLL}}$ at the latent variables, preventing it from modifying the values of $z$ and affecting the parameters of the encoder. We demonstrate the issues with regularized GLF in Section 4.2.1.

## 4 EXPERIMENTS

To demonstrate the performance of our method, we present both quantitative and qualitative evaluations on four commonly used datasets for generative models: MNIST (Lecun, 2010), Fashion MNIST (Xiao et al., 2017), CIFAR-10 (Krizhevsky et al., 2009) and CelebA (Liu et al., 2015). Throughout the experiments, we use 20-dimensional latent variables for MNIST and Fashion MNIST, and 64-dimensional latent variables for CIFAR-10 and CelebA.

(Lucic et al., 2018) adopted a common network architecture based on InfoGAN (Chen et al., 2016a) to evaluate GANs. In order to make fair comparisons without designing arbitrarily large networks to achieve better performance, we use the generator architecture of InfoGAN as our decoder's

architecture, and the encoder is set symmetric to the decoder. For details of the AE network structures, see Appendix A. For the flow applied on the latent variables, we use 4 affine coupling blocks as shown in Figure 1b, where each block contains 3 fully connected layers each with $k$ hidden units. For MNIST and Fashion MNIST, $k = 64$, while for CIFAR-10 and CelebA, $k = 256$. Note that the flow only adds a small parameter overhead on the auto-encoder (less than $3\%$).

## 4.1 METRICS

Estimated test data log likelihood is a popular metric to evaluate models based on VAEs. It is not trivial to estimate the log likelihood obtained from GLF, as it uses an deterministic auto-encoder. More importantly, as shown in (Grover et al., 2018; Theis et al., 2015), likelihood is not well correlated with sample quality. We use the Fréchet Inception Distance (FID) (Heusel et al., 2017) as a metric for image generation quality. FID is computed by first extracting features of a set of real images $x$ and a set of generated images $g$ from an intermediate layer of the Inception network (Szegedy et al., 2015). Each set of features is fitted with a Gaussian distribution, yielding means $\mu_x$, $\mu_g$ and co-variances matrices $\Sigma_x$, $\Sigma_g$. The FID score is defined to be the Fréchet distance between these two Gaussians:

$$\text{FID}(x, g) = \|\mu_x - \mu_g\|_2^2 + \text{Tr}\left(\Sigma_x + \Sigma_g - 2\left(\Sigma_x \Sigma_g\right)^{\frac{1}{2}}\right)$$

It is claimed that the FID score is sensitive to mode collapse and correlates well with human perception of generator quality (Lucic et al., 2018). Recently, (Sajjadi et al., 2018) proposed using Precision and Recall for Distributions (PRD) which can assess both the quality and diversity of generated samples. We also include PRD in our studies.

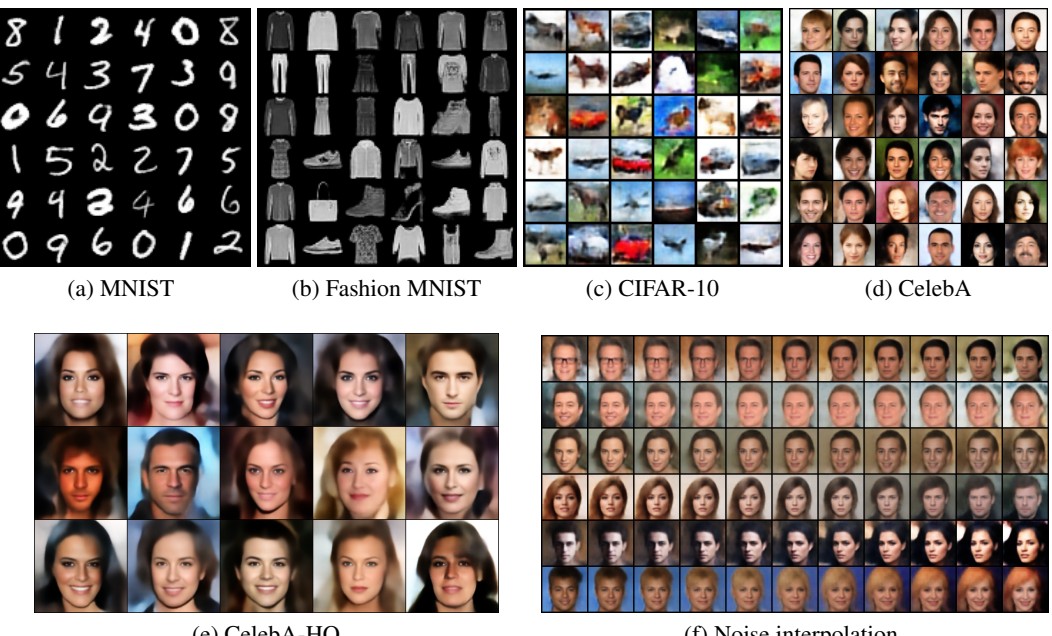

|  (a) MNIST | (b) Fashion MNIST | (c) CIFAR-10 | (d) CelebA |

| (e) CelebA-HQ | (f) Noise interpolation |

Figure 2: (a)-(e): Randomly generated samples from our method trained on different datasets. (f): Random noise interpolation on CelebA.

## 4.2 RESULTS

Table 1 summarizes the main results of this work. We compare the FID scores obtained by GLF with the scores of the VAE baseline and several existing AE based models that are claimed to produce high quality samples. Instead of directly citing their reported results, we re-ran the experiments because we want to evaluate them under standardized settings so that all models adopt the same AE architectures, latent dimensions and image pre-processing. We report the results of VAE+flow prior/posterior with

Table 1: FID scores obtained from different models. For our reported results, we executed 10 independent trials and report the mean and standard deviation of the FID scores. Each trail is computing the FID between 10k generated images and 10k real images.

|  | MNIST | Fashion | CIFAR-10 | CelebA |
|---|---|---|---|---|
| VAE | $28.2 \pm 0.3$ | $57.5 \pm 0.4$ | $142.5 \pm 0.6$ | $71.0 \pm 0.5$ |
| WAE-GAN | $12.4 \pm 0.2$ | $31.5 \pm 0.4$ | $93.1 \pm 0.5$ | $66.5 \pm 0.7$ |
| Two-Stage VAE | $10.9 \pm 0.7$ | $26.1 \pm 0.9$ | $96.1 \pm 0.9$ | $65.2 \pm 0.8$ |
| RAE + GMM | $10.8 \pm 0.1$ | $25.1 \pm 0.2$ | $91.6 \pm 0.6$ | $57.8 \pm 0.4$ |
| VAE+flow prior | $28.3 \pm 0.2$ | $51.8 \pm 0.3$ | $110.4 \pm 0.5$ | $54.3 \pm 0.3$ |
| VAE+flow posterior | $26.7 \pm 0.3$ | $55.1 \pm 0.3$ | $143.6 \pm 0.8$ | $67.9 \pm 0.3$ |
| GLF (ours) | $\mathbf{8.2} \pm 0.1$ | $\mathbf{21.3} \pm 0.2$ | $\mathbf{88.3} \pm 0.4$ | $\mathbf{53.2} \pm 0.2$ |
| GLANN with perceptual loss | $8.6 \pm 0.1$ | $13.0 \pm 0.1$ | $46.5 \pm 0.2$ | $46.3 \pm 0.1$ |
| GLF+perceptual loss (ours) | $\mathbf{5.8} \pm 0.1$ | $\mathbf{10.3} \pm 0.1$ | $\mathbf{44.6} \pm 0.3$ | $\mathbf{41.8} \pm 0.2$ |

$\beta = 1$. For other methods, we largely follow their proposed experimental settings. Details of each experiment are presented in Appendix B.

Note that the authors of WAE propose two variants, namely WAE-GAN and WAE-MMD. We only report the results of WAE-GAN, as we found it consistently outperforms WAE-MMD. Note also that, GLANN (Hoshen et al., 2019) obtains impressive FID scores, but it uses perceptual loss (Johnson et al., 2016) as the reconstruction loss, while other models use MSE loss. The perceptual loss is obtained by feeding both training images and reconstructed images into a pre-trained network such as VGG (Simonyan & Zisserman, 2014), and computing the $L_1$ distance between some of the intermediate layers' activation. We also train our method with perceptual loss and compare with GLANN in the last two rows of Table 1.

As shown in Table 1, our method obtains significantly lower FID scores than competing AE based models across all four datasets. In particular, GLF greatly outperforms VAE+flow prior with the default setting of $\beta = 1$. A more detailed analysis and comparison between the two methods will be done in Section 4.2.1. We also confirm that VAE+flow posterior cannot improve generation quality. Perhaps the competing model with the closest performances to ours is RAE+GMM, which shares some similarity with GLF in that both methods fit the density of the latent variables of an AE explicitly. To compare our method with GANs, we also include the results from (Lucic et al., 2018) in Appendix D. In (Lucic et al., 2018), the authors conduct standardized and comprehensive evaluations of representative GAN models with large-scale hyper-parameter searches, and therefore, their results can serve as a strong baseline. The results indicate that our method's generation quality is competitive with that of carefully tuned GANs.

In Table 3, Appendix C, we present the Precision and Recall scores of our method and several competing methods. As shown in the table, GLF obtains state-of-the-art Precision and Recall scores across all datasets, indicating that our method outperforms competing methods in terms of both sample quality and diversity.

Some qualitative results are shown in Figure 2. Besides samples of the datasets used for quantitative evaluation, samples of CelebA-HQ (Karras et al., 2017) with the larger size of $256 \times 256$ are also included in Figure 2e to show our method's ability to scale up to images with higher resolution. Qualitative results show that our model can generate sharp and diverse samples in each dataset. In Figure 2f, we show CelebA images generated by linearly interpolating two sampled random noise vectors. The smooth and natural transition shows that our model can generate samples that have not been seen during training. To provide further evidence that our model does not overfit or 'memorize' the training set, we show nearest neighbors in the training set for some generated samples in Appendix G. For more qualitative results, including samples from models trained with perceptual loss, see Appendix H. We observe that samples from models trained with perceptual loss have higher quality.

### 4.2.1 COMPARISONS: GLF VS. REGULARIZED GLF AND VAE+FLOW PRIOR.

As discussed in section 1 and section 3.2, we underline the novel finding regarding the relation between the weight on the reconstruction loss and the sample quality of VAEs with flow prior. In this

section, we present detailed experiments on this relation. We train VAEs+flow prior on CIFAR-10 for different choices of $\beta$, plus one with a learnable $\beta$ (Dai & Wipf, 2019). We record the progression of FID scores of these models in Figure 3a. In Figure 3b, we plot the entropy term, which is the last term in equation 3, the objective of VAE+flow prior. The entropy is expressed as $-\sum_{j=1}^{d} \log(\sigma_j)/2$, where $\sigma_j$ is the standard deviation of the approximate posterior on the $j^{th}$ latent variable. Higher entropy means that the latent variables have lower variances. In Figure 3c, we plot the NLL loss. We omit the results for $\beta = 1$ because the obtained FID scores are too high to fit the scale of the plot. Settings for the experiments in this subsection can be found in Appendix B.6.

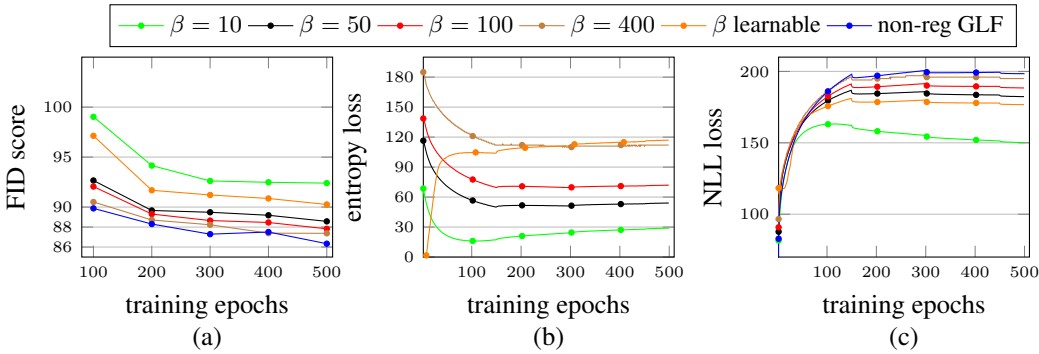

Figure 3: (a) Record of FID scores on CIFAR-10 for VAEs+flow prior with different values of $\beta$ and GLF. (b) Record of entropy losses for corresponding models. (c) Record of NLL losses for corresponding models.

From Figure 3a, we clearly observe the trend that the generation quality measured by FID scores improves as $\beta$ increases. We also observe that as $\beta$ increases, the performance gap between VAE+flow prior and GLF closes, indicating that GLF captures the limiting behavior of VAE+flow prior. We also find that learnable $\beta$ is not effective, probably due to the relatively small values of $\beta$ at the early stages of training. When $\beta$ is large, as indicated by Figure 3b, the posterior variances of VAEs become very small, so that effectively we are training an AE. For example, as shown in Figure 3b, when $\beta = 400$, the corresponding average posterior variance is around $10^{-4}$. This motivates us to use a deterministic auto-encoder in GLF, which as we have said above can be seen as the vanishing observational variance limit of VAE+flow prior. It is important to note that the relation between $\beta$ and generation quality only exists for VAEs with a trainable prior (such as normalizing flow), as we verify empirically that increasing $\beta$ on plain VAEs leads to worse FID scores.

As discussed in section 3.3.1, training regularized GLF is unstable because of the degeneracy of the latent variables driven by the NLL loss. We empirically study the effect of latent regularization as a function of $\beta$ and present results in Figure 4. For low values of $\beta = 1$ and 10, the NLL loss completely dominates the learning signal and the reconstruction loss quickly diverges, therefore we omit them in the plot. For larger values of $\beta = 50, 100, 400$ we observe that the NLL loss decreases to a negative value of very large magnitude, and although overall performance is reasonable, it oscillates quite strongly as training proceeds. In contrast, for GLF, where the flow does not modify $z$, the NLL loss does not degenerate, resulting in stable improvements of FID scores as training progress.

In contrast to regularized GLF, which uses a deterministic encoder, no degeneracy in the latent variables is observed for VAE+flow prior, thanks to the noise introduced in the stochastic encoder and the corresponding entropy term. Indeed, Figure 3c shows that the training of VAE+flow prior does not over-fit the NLL loss, as opposed to regularized GLF where severe over-fitting to NLL loss occurs as shown in Figure 4c. Comparing Figure 3a and 4a, we observe that unlike regularized GLF, VAE+flow prior does not suffer from divergence or fluctuations in FID scores, even with relatively small $\beta$. In summary, the results of FID scores show that regularized GLF is unstable, while as $\beta$ increases, the performance of VAE+flow prior converges to that of GLF. Note that although GLF only slightly outperforms VAE+flow prior even when $\beta$ is very large, it has the advantage that there is no need to tune $\beta$.

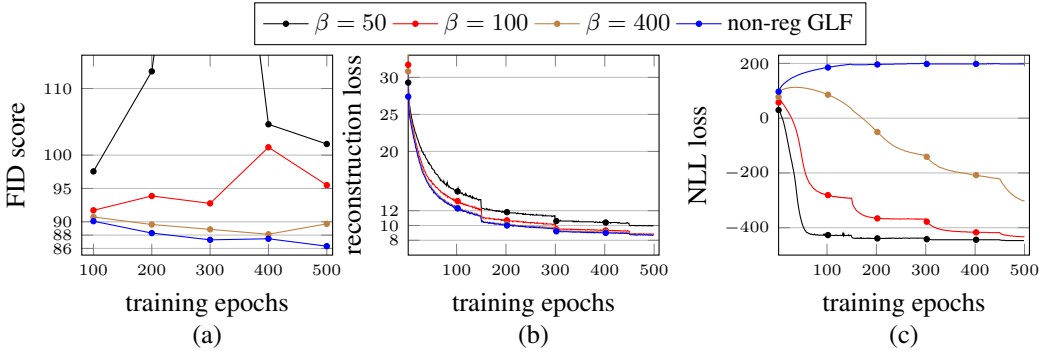

Figure 4: (a) Record of FID scores on CIFAR-10 for regularized GLF with different values of $\beta$ and GLF. $\beta = 1$ and 10 are omitted because they lead to divergence in the reconstruction loss. (b) Record of reconstruction loss for the corresponding models. (c) Record of NLL loss for the corresponding models.

### 4.3 TRAINING TIME

Besides better performance, our method also has the advantage of faster convergence among competing methods such as GLANN and Two-stage VAE. In Table 5, Appendix F, we compare the number of training epochs to obtain the FID scores in Table 1. We also compare the per epoch training clock time in Table 6, Appendix F. The combined results indicate that GLF requires much less training time while generating samples with higher quality.

## 5 CONCLUSION

In this paper, we introduce Generative Latent Flow, a novel generative model which uses an auto-encoder to learn a latent space from training data and a normalizing flow to match the distribution of the latent variables with the prior. Under standardized evaluations, our model achieves state-of-the-art results in image generation quality and diversity among several recently proposed auto-encoder based models. While we are not claiming that our GLF model is superior to GANs, we do believe that it opens the door to realizing the potential of AE based models to produce high quality samples just as GANs do. Our proposed model is motivated by our novel finding on the relation between large reconstruction weight and generation quality of VAEs with normalizing flow prior. The finding itself is important, as it can potentially motivate future work to study the trade-off between reconstruction and density matching in the objective of VAEs with learnable priors.

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

Table 2: Network structure for auto-encoder based on InfoGAN

| Encoder | Decoder |
| --- | --- |
| Input $x$ | Input $z$ |
| $4 \times 4$ Conv$_{64}$, ReLU | FC nz $\rightarrow 1024$, BN, ReLU |
| $4 \times 4$ Conv$_{128}$, BN, ReLU | FC $1024 \rightarrow 128 \times M \times M$, BN, ReLU |
| Flatten, FC $128 \times M \times M \rightarrow 1024$, BN, ReLU | $4 \times 4$ Deconv$_{64}$, BN, ReLU |
| FC $1024 \rightarrow$ nz | $4 \times 4$ Deconv$_{128}$, Sigmoid |

## A    NETWORK ARCHITECTURES

In this section we provide Table 2 that summarizes the auto-encoder network structure. The network structure is adopted from InfoGAN(Chen et al., 2016a), and the difference between the networks we used for each dataset is the size of the fully connected layers, which depends on the size of the image. All convolution and deconvolution layers have stride $= 2$ and padding $= 1$ to ensure the spatial dimension decreases/increases by a factor of 2. $M$ is simply the size of an input image divided by 4. Specifically, for MNIST and Fashion MNIST, $M = 7$; for CIFAR-10, $M = 8$; for CelebA, $M = 16$. BN stands for batch normalization.

For VAEs, the final FC layer of the encoder will have doubled output size to return both the mean and standard deviation of latent variables.

## B    EXPERIMENT SETTINGS

In this section, we present the details of our experimental settings for results in Table 1. Since the settings for MNIST and Fashion MNIST are the same, we only mention MNIST for simplicity. For GLANN, we directly cite the results from (Hoshen et al., 2019), as their experimental settings is very similar to ours.

We use the original images in the training sets for MNIST, Fashion MNIST and CIFAR-10. For CelebA, we follow the same pre-processing as in (Lucic et al., 2018): center crop to $160 \times 160$ and then resize to $64 \times 64$. We normalize the pixel values to $[0, 1]$, without adding noise to pixels (i.e, no de-quantization).

### B.1    SETTINGS FOR TRAINING GLF

For all datasets (except CelebA-HQ), we use batch size 256 and Adam (Kingma & Ba, 2014) optimizer with initial learning rate $10^{-3}$ for the parameters of both the AE and the flow. We add a weight decay $2 \times 10^{-5}$ to the optimizer for the flow. For MNIST, we train our model for 100 epochs, with learning rate decaying by a factor of 2 after 50 epochs. For CIFAR-10, we train our model for 200 epochs, with the learning rate decaying by a factor of 2 every 50 epochs. For CelebA, we train our model for 40 epochs with no learning rate decay.

For GLF with perceptual loss, we compute the perceptual loss as suggested in (Hoshen & Wolf, 2018). See `https://github.com/facebookresearch/NAM/blob/master/code/perceptual_loss.py` for their implementation. Other settings are the same.

For CelebA-HQ dataset, we adopt our AE network structure based on DCGAN (Radford et al., 2015). Note that this is a relatively simple network for high resolution imgaes. We use batch size 64, with initial learning rate $10^{-3}$ for both the AE and the flow. We train our model for 60 epochs, with learning rate decaying by a factor of 2 after 40 epochs.

### B.2    SETTINGS FOR TRAINING VAES AND VAE VARIANTS

We adopt common settings for our reported results of VAE, VAE+flow prior and VAE+flow posterior. We use $\beta = 1$ for all three VAE variants. We still use batch size 256, and Adam optimizer with initial learning rate $10^{-3}$ for both the VAE and the flow, if applicable. We find VAEs need longer time to converge, so we double the training epochs. We train MNIST for 200 epochs, with learning rate

decaying by a factor of 2 after 100 epochs. We train CIFAR-10 for 400 epochs, with the learning rate decaying by a factor of 2 every 100 epochs. We train CelebA for 80 epochs with learning rate decaying by a factor of 2 after 40 epochs.

### B.3 SETTINGS FOR TRAINING WAE-GAN

We follow the settings introduced in the original WAE paper(Tolstikhin et al., 2017). The adversary in WAE-GAN has the following architecture:

$$z \in \mathcal{R}^d \to \text{FC}_{512} \to \text{ReLU}$$
$$\to \text{FC}_{512} \to \text{ReLU}$$
$$\to \text{FC}_{512} \to \text{ReLU}$$
$$\to \text{FC}_{512} \to \text{ReLU} \to \text{FC}_1$$

where $d$ is the dimension of the latent variables.

WAE has two major hyper-parameters: $\lambda$ that controls the weight coefficient of the adversarial regularizer, and $\sigma^2$ which is the variance of the prior. Batch size is 100 for all datasets. For MNIST, $\lambda = 10$ and $\sigma^2 = 1$, and the model is trained for 100 epochs. The initial learning rate is $10^{-3}$ for the AE and $5 \times 10^{-4}$ for the adversary. After 30 epochs both learning rates decreased both by factor of 2, and after first 50 epochs further by factor of 5. For CIFAR, $\lambda = 10$ and $\sigma^2 = 1$ and the model is trained for 200 epochs. The initial learning rates are the same as training MNIST, and the learning rate decays by a factor of 2 after first 60 epochs, and further by a factor of 5 after 120 epochs. For CelebA, $\lambda = 1$ and $\sigma^2 = 2$. The model is trained for 55 epochs. The initial learning rate is $3 \times 10^{-4}$ for the AE and $10^{-3}$ for the adversary. Both learning rates decays by factor of 2 after 30 epochs, further by factor of 5 after 50 first epochs.

### B.4 SETTINGS FOR TRAINING TWO STAGE VAE

We adopt the settings in the original paper (Dai & Wipf, 2019). For all datasets, the batch size is set to be 64, and the initial learning rate for both the first and the second is $10^{-4}$. For MNIST, the first VAE is trained for 400 epochs, with learning rate halved every 150 epochs; the second VAE is trained for 800 epochs with learning rate halved every 300 epochs. For CIFAR-10, 1000 and 2000 epochs are trained for the two VAEs respectively, and the learning rates are halved every 300 and 600 epochs for the two stages. For CelebA, 120 and 300 epochs are trained for the two VAEs respectively, and the learning rates are halved every 48 and 120 epochs for the two stages.

**Explaining the discrepancy between our reported results and the results in the original paper:** The original Two stage VAE paper adopts similar settings with our experiments, but we observe large discrepancies on the results of CIFAR-10 and CelebA. After carefully reviewing their published codes, we find that there is an issue in their FID score computation particularly for CIFAR-10 dataset. Specifically, the true images used for computing the FID on CIFAR-10 is obtained from saving the original data file in .jpg format and reading them back, and the saving will cause some errors in pixel values. After fixing this issue, we re-ran their published codes and obtained similar results as we reported. We also run through their original FID computation protocol using samples from our models, and we obtain scores around 65. For CelebA, one particular detail worth noting is that, (Dai & Wipf, 2019) applies $128 \times 128$ center-crop before re-sizing on CelebA, while $160 \times 160$ center-crop is used in our evaluations. With smaller center-crops the human faces occupy a larger portion of the image with less background, making the generative modeling easier.

### B.5 SETTINGS FOR TRAINING RAE+GMM

The settings of batch size, learning rate scheduling and number of epochs for training RAE are the same as those of GLF. The objective of the RAE is reconstruction loss plus a penalty on the norm of the latent variable. Since the author does not report their choices for the penalty coefficient $\gamma$, we search over $\gamma \in 0.1, 0.5, 1, 2$, and we find that $\beta = 0.5$ leads to the best overall performances, and therefore we let $\gamma = 0.5$. After training the RAE, we fit a 10-component Gaussian mixture distribution on the latent variables.

## B.6 SETTINGS FOR EXPERIMENTS IN SECTION 4.2.1

For all experiments in Section 4.2.1, we use batch size 256 and initial learning rate $10^{-3}$ for both AE and flow. We train all models for 500 epochs with learning rates decaying by a factor of 2 every 150 epochs.

## C PRECISION AND RECALL

In this section, we report the precision and recall (PRD) evaluation of samples on each dataset in Table 3. We include WAE-GAN, Two-stage VAE, RAE+GMM and GLANN (with perceptual loss) for comparisons. As the case of FID scores, for GLANN, we directly cite their reported results, and we compute the results ofr other models. We report the PRD of models trained under the settings introduced in Appendix B.

The two numbers in each entry are $F_8, F_{\frac{1}{8}}$ that capture recall and precision, respectively. See (Sajjadi et al., 2018) for more details. Higher numbers are better.

Table 3: Evaluation of sample quality by precision/recall.

|  | MNIST | Fashion | CIFAR-10 | CelebA |
|---|---|---|---|---|
| WAE-GAN | (0.978, 0.956) | (0.901, 0.837) | (0.414, 0.723) | (0.501, 0.512) |
| Two-stage VAE | (0.982, 0.977) | (**0.937**, 0.845) | (0.382, 0.669) | (0.452, 0.558) |
| RAE+GMM | (**0.988**, 0.971) | (0.922, 0.924) | (0.370, 0.733) | (0.333, 0.445) |
| GLF (ours) | (0.982, **0.985**) | (0.932, **0.926**) | (**0.485, 0.767**) | (**0.542, 0.618**) |
| GLANN+perceptual loss | (0.971, 0.979) | (0.985, 0.963) | (**0.860**, 0.825) | (0.574, 0.681) |
| GLF+perceptual loss (ours) | (**0.990, 0.992**) | (**0.987, 0.980**) | (0.765, **0.845**) | (**0.760, 0.778**) |

## D COMPARISON WITH GANS

In Table 4 we combine our reported results of AE based models and the FID scores of GANs cited from (Lucic et al., 2018).

Table 4: FID score comparisons of GANs and various AE based models

|  | MNIST | Fashion | CIFAR-10 | CelebA |
|---|---|---|---|---|
| MM GAN | $9.8 \pm 0.9$ | $29.6 \pm 1.6$ | $72.7 \pm 3.6$ | $65.6 \pm 4.2$ |
| NS GAN | $6.8 \pm 0.5$ | $26.5 \pm 1.6$ | $58.5 \pm 1.9$ | $55.0 \pm 3.3$ |
| LSGAN | $7.8 \pm 0.6$ | $30.7 \pm 2.2$ | $87.1 \pm 47.5$ | $53.9 \pm 2.8$ |
| WGAN | $6.7 \pm 0.4$ | $21.5 \pm 1.6$ | $55.2 \pm 2.3$ | $41.3 \pm 2.0$ |
| WGAN GP | $20.3 \pm 5.0$ | $24.5 \pm 2.1$ | $55.8 \pm 0.9$ | $30.3 \pm 1.0$ |
| DRAGAN | $7.6 \pm 0.4$ | $27.7 \pm 1.2$ | $69.8 \pm 2.0$ | $42.3 \pm 3.0$ |
| BEGAN | $13.1 \pm 1.0$ | $22.9 \pm 0.9$ | $71.4 \pm 1.6$ | $38.9 \pm 0.9$ |
| VAE | $28.2 \pm 0.3$ | $57.5 \pm 0.4$ | $142.5 \pm 0.6$ | $71.0 \pm 0.5$ |
| WAE-GAN | $12.4 \pm 0.2$ | $31.5 \pm 0.4$ | $93.1 \pm 0.5$ | $66.5 \pm 0.7$ |
| Two-Stage VAE | $10.9 \pm 0.7$ | $26.1 \pm 0.9$ | $96.1 \pm 0.9$ | $65.2 \pm 0.8$ |
| RAE + GMM | $10.8 \pm 0.1$ | $25.1 \pm 0.2$ | $91.6 \pm 0.6$ | $57.8 \pm 0.4$ |
| GLANN (with perceptual loss) | $8.6 \pm 0.1$ | $13.0 \pm 0.1$ | $46.5 \pm 0.2$ | $46.3 \pm 0.1$ |
| VAE+flow prior | $28.3 \pm 0.2$ | $51.8 \pm 0.3$ | $110.4 \pm 0.5$ | $54.3 \pm 0.3$ |
| VAE+flow posterior | $26.7 \pm 0.3$ | $55.1 \pm 0.3$ | $143.6 \pm 0.8$ | $67.9 \pm 0.3$ |
| GLF (ours) | $8.2 \pm 0.1$ | $21.3 \pm 0.2$ | $88.3 \pm 0.4$ | $53.2 \pm 0.2$ |
| GLF+perceptual loss (ours) | $5.8 \pm 0.1$ | $10.3 \pm 0.1$ | $44.6 \pm 0.3$ | $41.8 \pm 0.2$ |

## E    ISSUES WITH LATENT REGULARIZATION

In this appendix section, we present the plots of FID scores, reconstruction loss and NLL loss of regularized GLF. Related results are discussed in Section 4.2.1.

## F    TRAINING TIME COMPARISONS

In Table 5, we report the number of training epochs of our method, two-stage VAE and GLANN. In Table 6, we report the clock training time per epoch of these methods. Note that for methods using perceptual loss, the per epoch training time is longer because VGG activations need to be computed. These two tables show that GLF needs much shorter training time than the two competing methods. In GLF, training the flow does not add much computational time due to the low dimensionality.

Table 5: Number of training epochs for Two-stage VAE, GLANN, and GLF

|                            | MNIST/Fashion | CIFAR-10  | CelebA  |
| -------------------------- | ------------- | --------- | ------- |
| Two-stage VAE First/Second | 400/800       | 1000/2000 | 120/300 |
| GLANN First/Second         | 500/50        | 500/50    | 500/50  |
| GLF                        | 100           | 200       | 40      |

Table 6: Per-epoch training time in seconds

|                           | MNIST/Fashion | CIFAR-10 | CelebA |
| ------------------------- | ------------- | -------- | ------ |
| 2-stage VAE 1st/2nd       | 5/2           | 6/2      | 60/28  |
| GLF                       | 10            | 13       | 108    |
| GLANN with perceptual loss | 14           | 16       | 292    |
| GLF with perceptual loss  | 16            | 19       | 343    |

## G    NEAREST NEIGHBORS OF GENERATED SAMPLES IN THE TRAINING SET

Quantitative measurements of sample quality, such as FID score and Precision/Recall can be minimized by letting the generative model memorize the training set. The smooth transition of noise interpolation shown in Figure 2 provides evidence that our model can generalize, i.e., it generate samples that have not be seen during training. Here we provide additional evidence showing that our generative model generalizes well.

We randomly generate some samples from the models trained on MNIST and CelebA datasets. Then we present the 5 nearest neighbors of each generated samples in the training set. The nearest neighbor is defined in terms of $L_2$ distance. Results are shown in Figure 5. By inspecting the figure, we find that we can easily differentiate each generated sample from the closest training data. This indicates that our model generalizes well.

## H    MORE QUALITATIVE RESULTS

In Figure 6, we show more samples of each dataset generated by GLF, using either MSE or perceptual loss as reconstruction loss. In Figure 7, we show samples of CelebA-HQ datasets from GLF trained with perceptual loss. In Figure 8, we show examples of interpolations between two randomly sampled noises on CelebA from GLF trained with perceptual loss.

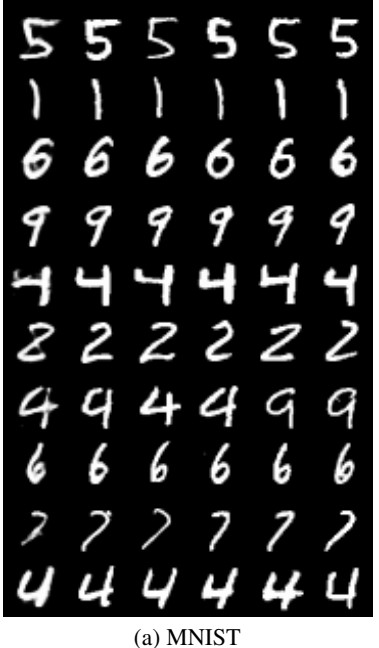

(a) MNIST

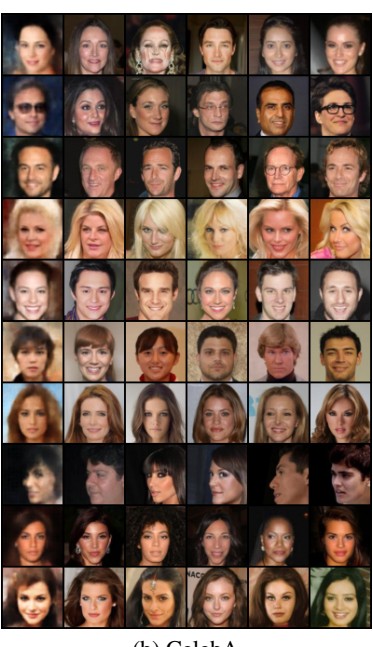

(b) CelebA

Figure 5: Some randomly generated samples are presented in the **leftmost** column in each picture. The other 5 columns of each picture show the top 5 nearest neighbors of the corresponding sample in the training set.

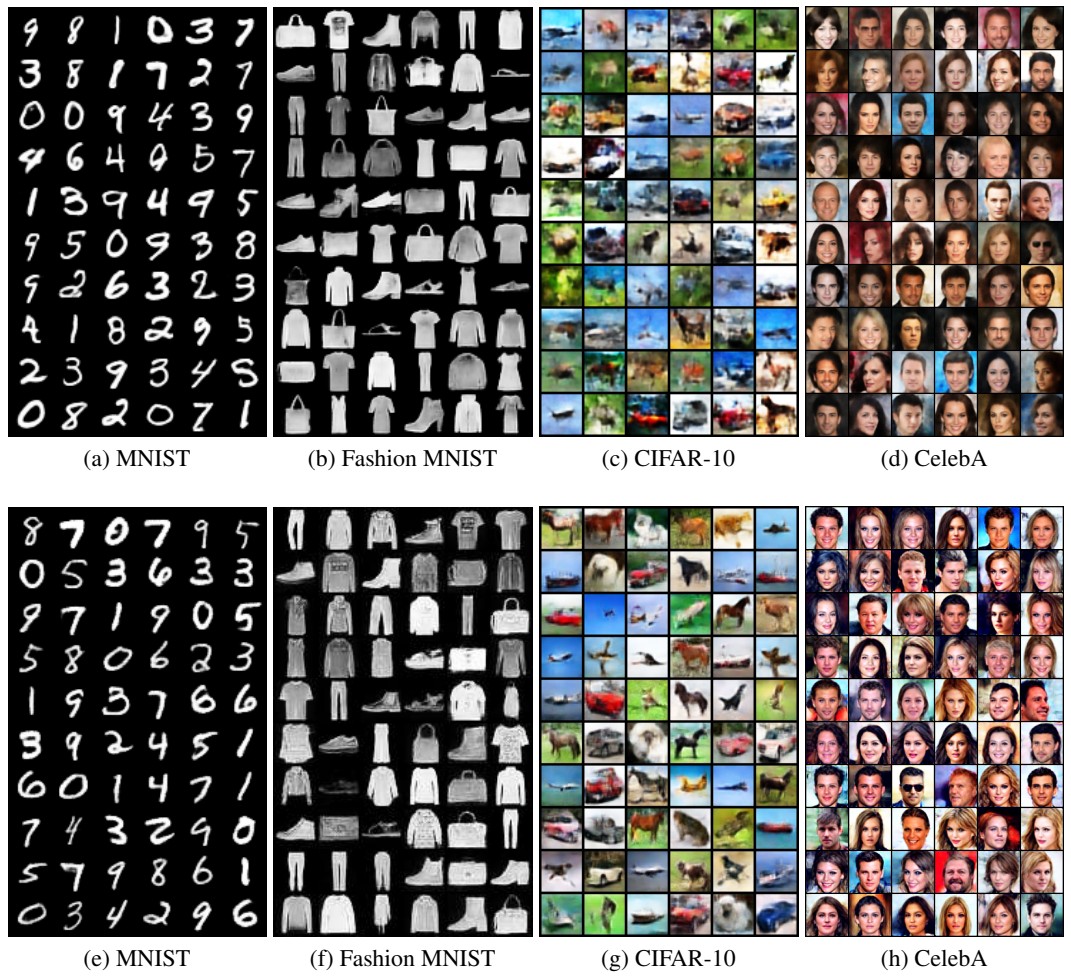

(a) MNIST    (b) Fashion MNIST    (c) CIFAR-10    (d) CelebA

(e) MNIST    (f) Fashion MNIST    (g) CIFAR-10    (h) CelebA

Figure 6: (a)-(d) Randomly generated samples from our method with MSE loss. (e)-(h) Randomly generated samples from our method with perceptual loss.

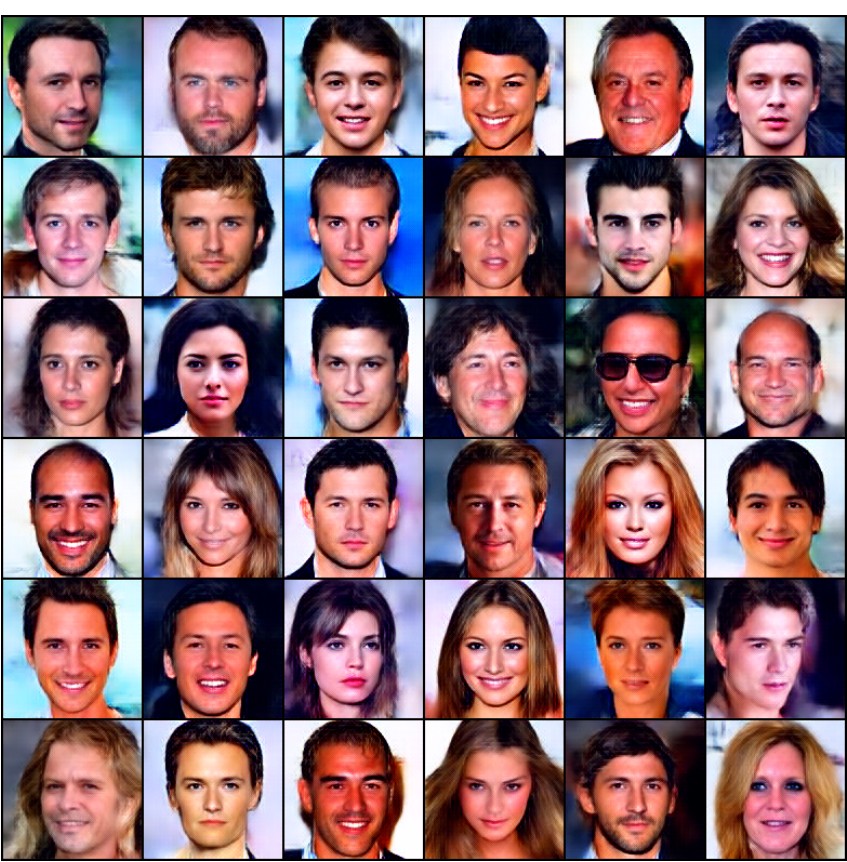

Figure 7: Randomly generated samples from our method with perceptual loss on CelebA-HQ dataset

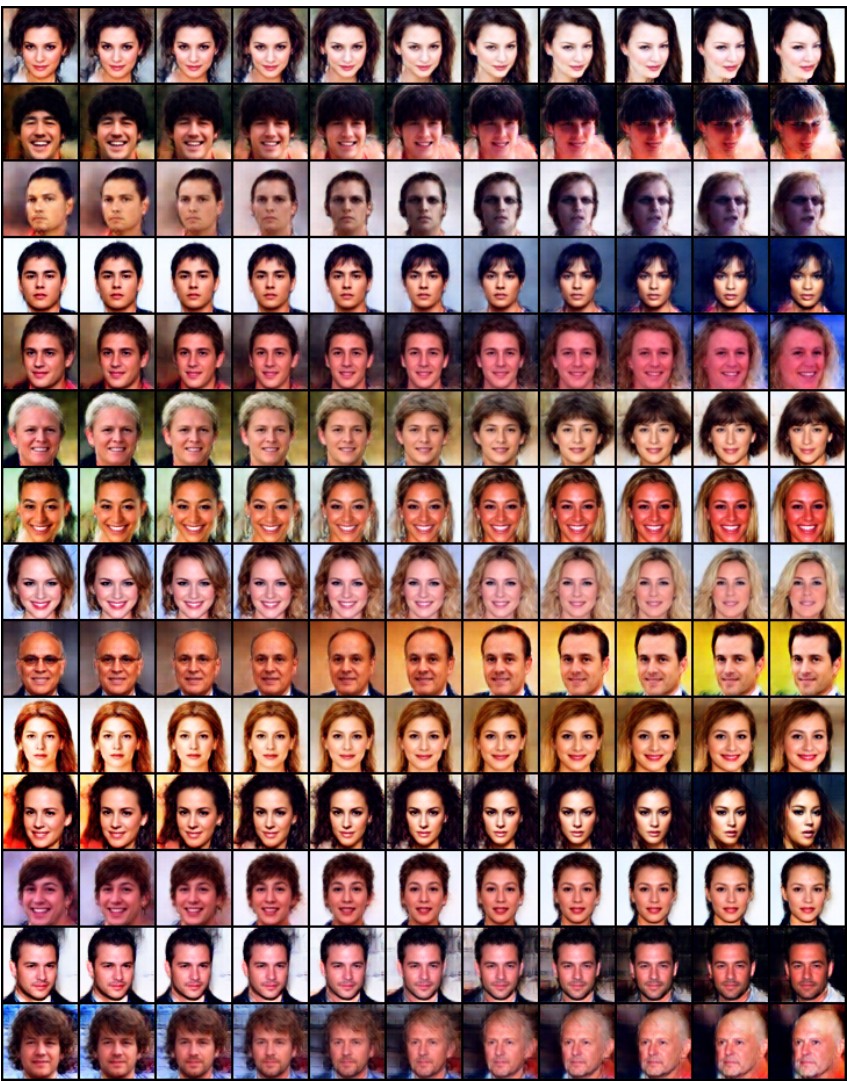

Figure 8: Noise interpolation on CelebA

