# OpenReview forum: "Generative Latent Flow"
_ICLR.cc/2020/Conference — Reject_

### Official Review · AnonReviewer3 · 2019-10-22
**Official Blind Review #3**

**Rating:** 3

**Review:**

Summary:
The authors propose a model that combines a simple Auto-Encoder (AE) together with a Normalizing Flow (NF) model, such that to derive a generative model. In particular, the AE is used to learn a low-dimensional representation of the given data in a latent space. Then, a NF model learns under a maximum likelihood principle, the distribution of these latent codes by applying an invertible transformation on a easy to sample distribution.

General Comments:

The paper is ok written, but some parts probably can be improved (see comment #5). As regards the proposed idea of the paper, I think that it is closer to an engineering practical approach than a consistent modeling choice. In particular:

1. With the current modeling the likelihood p(x|z) is not defined, which means that the density of the p(x) can not be evaluated. So this should be considered as an explicit or implicit generative model? In that case the comparison with explicit density models where we can evaluate the p(x) is a bit unfair. Otherwise, the test log-likelihood should be provided.

2. Learning the prior is already a debatable choice. However, in the proposed idea I have the feeling that there is a strong overfiting issue. Since the NF model is trained in a maximum likelihood principle, it will try to put all the mass from the simple distribution p(e) on the training latent codes z_i. Consequently, I am a bit sceptical as regards the generalization power of the generative model. Does the latent distribution learn something meaningful (an illustration could have been informative) or just how to re-generate the training data?

3. The authors claim that the training is end-to-end. However, I think that the model which performs better is actually a 2 stage training model. More specifically, due the sg[.], the L_NLL term does not have any influence in the L_recon term. Therefore, the AE is trained independently, and simply the NF at every step "follows" and tries to capture the latent (empirical) distribution of the encoded training data.

4. From the experiments is argued that the proposed model provides better samples than the competitive methods. I have the feeling that the generated samples are basically very similar to the training samples. Because the learned prior essentially learns to generate the latent codes of the training data. Does the model generalize i.e. can it generate samples that have not be seen during training?

5. I think that the first paragraph of Sec. 2 and some parts of the next paragraph need improvement. Also, how the decoder implies the distribution \tilde{p}(z) in the latent space? I would expect the encoder to be responsible for the latent distribution. Moreover, in the classic VAE the KL divergence is used between the approximate posterior q(z|x) and p(z), while the KL between the aggregated posterior q(z) and p(z) is not the default choice, and usually, is not straightforward to optimize.

In general, I think that the proposed model is a rather good practical approach, but probably not a very well defined modeling choice. As a practical approach, the experiments is most of the times the only way to support the argued behavior. The conducted experiments mainly focus on the FID score. However, I think that it would have been interesting to include examples that show the latent distribution and why is this better from other models, for example less regularized from the p(z) in VAE. Also, another crucial issue is the level of overfiting the current approach might have, because learning the prior implies this behaviour. Does the generated samples cover only the training distribution or can it generalizes to unseen test samples?

**Experience Assessment:**

I have published one or two papers in this area.

**Review Assessment: Checking Correctness Of Derivations And Theory:**

I assessed the sensibility of the derivations and theory.

**Review Assessment: Checking Correctness Of Experiments:**

I assessed the sensibility of the experiments.

**Review Assessment: Thoroughness In Paper Reading:**

I read the paper at least twice and used my best judgement in assessing the paper.

---

> ### Author Response · Authors · 2019-11-10
> **Replies to comments**
>
> We thank the reviewer for insightful comments. We will re-organize section 2 and 3, as suggested by the reviewer.
>
> First of all, we would like to address the concern that our work is just an engineering practical approach. Our work not only proposes a simple yet effective generative model, but also has some important implications. Basically, we make the novel finding that increasing the weight ($\beta$) of reconstruction in the objective of VAEs with learnable priors improves generation quality, and this finding motivates us to propose GLF by considering the case when $\beta$ is extremely large. The relation between $\beta$ and generation quality is important as it can motivate future works to study the trade-off between reconstruction and density matching (think about beta-VAE, which uses large weight on KL). We will modify the paper to emphasize our contributions.
>
> 1. When using deterministic AE, the likelihood cannot be easily estimated. However,. in Table 1, most AE based models that obtain good generation performances, such as WAE, RAE+GMM and GLANN, use deterministic latent variables, so their likelihoods cannot be computed neither, so the comparison is fair. Moreover, our model can be seen as the vanishing noise limit of VAE+flow prior, whose likelihood can be estimated. However, this is not the main focus. Actually, increasing $\beta$ improves generation quality but has negative effect on likelihood.
>
> Both 2 and 4 are on the generalization of our model. Indeed, both metrics in our paper (FID and PRD) can be minimized by memorizing the training set, and currently we cannot find a well accepted quantitative metric for generalization. Nevertheless, some experiments can provide evidence for generalization. In the paper, we already present CelebA samples obtained from NOISE INTERPOLATION (sample two noise vectors, and generate samples using the linear interpolation of these two noise vectors) in Figure 2(f) (and additional results in appendix).  We observe that the each generated sample along the interpolation is realistic, and the transition from one sample to another is smooth. This provides strong evidence that the data distribution we fit has no 'holes'. In other words, our model does not simply memorize the training set. In the updated version, we will show nearest neighbors in training set of some generated samples. The generated samples are visually different from their nearest training data.
>
> 3. Indeed, as we mentioned in section 3 of the paper, AE and flow are trained independently, and GLF can be trained in either one-stage or two-stage. 'One-stage' just means that we do not have to train AE and flow separately, as the flow can capture the latent distribution during training. In practice, we do not observe two-stage training can improve the performance. We use one-stage training because VAE+flow prior is trained jointly, and our model is motivated by VAE+flow prior. We will make this point clearer in revised paper, and to avoid confusion, we will remove terms related to 'single-stage training'.
>
> 5. We will improve the writing as suggested. For the question regarding $\tilde{p}(z)$, what we want to say is a general assumption of generative models, namely given decoder $G$, every $x$ in the data distribution can be written as $x = G(z)$ for some $z$ in latent space. No encoder is defined, and thus $z$ and $x$ are related through $G$. In the case of Auto-encoder, certainly the distribution of $z$ depends on encoder. We realize our expression will cause confusion, and since it is not a major point of the paper, we will remove it.
>
> For the question regarding posterior v.s aggregated posterior, we know that the VAE's objective contains $KL(q(z|x)||p(z))$. What we said in paper is, [1] shows that $$\mathbb{E}_{p_{\text {data}}(x)}[K L(q(z | x) \| p(z))]=I(x ; z)+K L(q(z) \| p(z)),$$ so VAE's objective indirectly penalizes $K L(q(z) \| p(z))$ ($I(x ; z)$ is non-negative). We want to emphasize $K L(q(z) \| p(z))$ because this term is important for any generative models with latent variables to generate good samples, and the main reason to use learnable prior is to reduce the mismatch between $q(z)$ and $p(z)$.
>
> The reviewer suggests to show the latent distribution of our model and why is this better than VAEs. Our latent variables are just the latent variables of deterministic Auto-encoders, and they are better than VAEs' latent variables because of lower reconstruction loss. There is always a trade off between reconstruction and KL for VAEs' latent variables, but in our case, the trade off can be overcome because the latent variables maximize its ability to reconstruct the data, and the match with prior is purely done by the flow.
>
> We hope our reply address all the concerns. We thank the reviewer again, and we will let you know when the new version is uploaded.
>
> [1]  ELBO surgery: yet another way to carve up the variational evidence lower bound. Hoffman & Johnson

---

### Official Review · AnonReviewer1 · 2019-10-23
**Official Blind Review #1**

**Rating:** 3

**Review:**

The paper proposes a deterministic Auto-Encoder with a trained marginal distribution over latent variables, p(z), to be able to sample from the model. For this purpose, the authors propose to use a flow-based model for p(z), and regularize the AE objective (i.e., MSE) with a cross-entropy between q(z) = 1/N \sum_n E(x_n) and p(z). In general, I find the idea quite interesting. The construction of the objective and motivation behind is rather clear. The experiments are rather convincing (however, the FID metric is subjective, but it's impossible to calculate the log-likelihood scores). The main disadvantage of the paper is its language. There are many typos and difficult to follow sentences. But besides that, the main ideas are well explained. Please find more specific remarks below.

Remarks
- The language in the paper is a bit off. There are sentences that sound very peculiar, e.g., "Deep generative models can be roughly classified into explicit and implicit models. The former class assumes explicit parametric specification of the distribution, whereas the latter does not."
The introduction should be re-written. Similarly, Section 2 is hard to follow.
There are places where a word or a punctuation mark is missing, for instance:
* The first line misses a word: "(...) on deep learning."
* First paragraph, Section 2: "This distribution is unknown and possibly The model also has a predefined prior distribution p(z) on Z."
* Section 3.2, below Eq. 2: "(...) assumptio (...)".
* Section 3.4, last paragraph: "Our work is differs in two ways (...)".

- The authors should refer to the following very relevant paper:
* Xu, H., Chen, W., Lai, J., Li, Z., Zhao, Y., & Pei, D. (2019). On the Necessity and Effectiveness of Learning the Prior of Variational Auto-Encoder. arXiv preprint arXiv:1905.13452.

- It is unclear how the authors dequantize image data that are typically represent as integers from 0 to 255 (or 0 and 1 in the binary case). The authors mention in Section 3.2 that they use the MSE loss for \mathcal{L}_{recon}. This corresponds to taking a Gaussian decoder in the VAE framework. However, the manner how the images are dequantized is extremely important to properly evaluate all results. For instance, if the authors add a uniform noise, then it highly influences the final quality of a model.

- I do not fully follow why the authors call the objective part for learning the marginal distribution over latents "the NLL loss". It is rather the cross entropy between q(z) = 1/N \sum_n E(x_n) and p(z). This follows from the notation and the description that we stop the gradient for E(.). I find it confusing.

- Figure 4 is extremely important for understanding why we should stop gradient. However, jumping between Section 4.2 and E is extremely annoying. The authors can use up to 10 pages, so adding a half of a page would not make a difference, but it would help a reader a lot.

- The main difference between the objective in Eq. 2 and the ELBO lies in considering a deterministic encoder and entails skipping the entropy term for the variational posterior. Could the authors comment why this is so important to choose a deterministic encoder? I can easily imagine taking q(z|x) = Normal(z | mu(z), a1), i.e., fixing the variance to some value a (e.g., a=1), that would result in Entropy[q] = const. Hence, we can skip entropy from the objective, but still we use a stochastic encoder.

===== AFTER REBUTTAL =====
I would like to thank the authors for their response and new version of the paper. I must admit that I had a very hard nut to crack. From one side, I really appreciate all the effort the authors put to improve the paper. However, on the other hand, I tend to agree with the reviewer #3 that the paper is interesting from the engineering perspective and it lacks novelty. Eventually, I decided to sustain my score. There are two reasons for that. First, more analysis of the model would be helpful. For instance, analyzing the latent space would be interesting (a suggestion from the reviewer #3). Second, considering a pure AE with a trainable flow-based prior is interesting, but it is also very limiting in the sense of obtaining nicely looking pictures instead of being able to provide (approximate) probability of an image. In my opinion, from the decision making perspective, having a probability of an object is much more important  than being able to generate a crisp image.

**Experience Assessment:**

I have published in this field for several years.

**Review Assessment: Checking Correctness Of Derivations And Theory:**

I assessed the sensibility of the derivations and theory.

**Review Assessment: Checking Correctness Of Experiments:**

I assessed the sensibility of the experiments.

**Review Assessment: Thoroughness In Paper Reading:**

I read the paper at least twice and used my best judgement in assessing the paper.

---

> ### Author Response · Authors · 2019-11-09
> **Sorry for the writing issues, and reply to your questions**
>
> We thank the reviewer for providing insightful comments. We apologize for the inconvenience caused by our issues in writing, such as typos and missing words. We will carefully check the whole paper and upload a new version soon. In particular, we will re-organize section 1,2 and 3 to make them easier to follow.
>
> In the general comments, the reviewer commented that the log-likelihood of our model cannot be computed. This is true, but 1. FID and PRD are widely accepted metric for generation quality, and FID is used in all competing models, many of which do not have computable likelihood neither, and 2. likelihood is not well correlated with generation quality as confirmed in [1,2].
>
> We thank the reviewer for pointing out the related work by Xu et al. We will cite it when discussing VAE+flow prior in our paper, but we do want to emphasize that we focus on completely different perspectives of the same model. This work merely focuses on the likelihood of data, whereas we focus on generation quality, leading to some novel findings.
>
> - De-quantization of image:
>
> For all datasets we used, we normalize the pixel values to $[0,1]$ and we do not add any noise to the normalized image. In other words, there is no de-quantization. We know that it is important to add small amount of noise to the data when training a normalizing flow that directly transforms the data distribution, but our flow is applied on the latent space of an AE, and usually it is unnecessary to de-quantize the image when training AEs. We add clarification on this in Appendix B.
>
> - The name of NLL loss.
>
> We call the term in objective NLL loss because the normalizing flow is trained by the criteria of maximizing the likelihood of $z$, which is equivalent to minimize the negative log-likelihood (NLL), i.e., min $\mathcal -E_{q(z)}[\log p_\theta (z)]$, where $q(z)=\frac{1}{N} \sum_n E(x_n)$. Just as commented by reviewer, this is exactly the cross entropy between $q(z)$ and $p_{\theta}(z)$. Both names are fine, we use NLL loss because this is widely used in the normalizing flow literature to emphasize the training objective is (negative) likelihood. Hope the explanation will resolve your confusion.
>
> - We will move Figure 4 to the main text. Thanks for the suggestion!
>
> - Deterministic encoder versus stochastic encoder with constant variance
>
> We use deterministic encoders instead of stochastic ones because of our motivation to derive the model. GLF is motivated by our novel finding on VAE+flow prior. Previous works on VAE+flow prior only focus on likelihood, and they do not consider tuning the weight of reconstruction loss (i.e., always set $\beta=1$). We find that when $\beta=1$, VAE+flow prior does not obtain good FID scores,  however, sample quality consistently improves as $\beta$ increases. Therefore, we think about the case when $\beta$ is extremely large. Obviously, in this case the VAE's posterior will have very small variance (because the entropy term has negligible contribution), and therefore we take the vanishing noise limit, which results in Auto-encoder. Using constant variance is another way to get rid of the entropy term, but we do not see particular reasons to use it rather than simply using deterministic Auto-encoder. The randomness of $z$ prevents the model from over-fitting the likelihood loss, as we discuss in section 4.2.1. However, the over-fitting problem can be avoided using stop gradient operation, as we do in our model. In addition, we run some experiments with constant variance Auto-encoder, and observe that constant variance leads to slightly worse performances than deterministic encoder.
>
> We want to emphasize that the most important point of our model is not replacing stochastic encoder with deterministic encoder. The main point is that, the combination of normalizing flow prior AND large weight on reconstruction loss (both constant variance and deterministic encoders can be seen as the limiting case of large reconstruction weight, as entropy loss is removed) leads to better sample quality. So, both deterministic encoder and constant variance encoder should work. We will re-organize section 3 to make our reasoning easier to follow.
>
> We thank the reviewer again, and we will let you know when the new version is uploaded.
>
> [1] A note on the evaluation of generative models. Theis et al.
> [2] Flow-GAN: Combining Maximum Likelihood and Adversarial Learning in Generative Models. Grover et al.

---

### Official Review · AnonReviewer2 · 2019-10-24
**Official Blind Review #2**

**Rating:** 6

**Review:**

The paper proposes a new model combining an auto-encoder (AE) and a normalising flow (NF). The model, Generative Latent Flow (GLF), uses the AE to map the inputs to a latent space, which is then transformed using the NF. The approach is intuitively beneficial in that the AE can reduce the dimensionality of the inputs such that the NF mapping becomes much faster, computationally. The proposed method is compared to related methods that use a variational AE (VAE) in combination with an NF, and the similarities are pointed out and studied empirically.
The authors compare the performance of GLF to a large number of competing methods, showing very competitive results. In particular, for $\beta = 1$, GLF significantly outperforms its VAE+flow prior sibling in terms of the Fréchet Inception Distance (FID) measuring the quality of the generated images.

The paper presents a well-motivated method, which is extensively and thoroughly evaluated. The experimental part is the paper's main strength, as the method itself is quite incremental (replacing a VAE with an AE). The authors do, however, spend considerable effort comparing the two versions of the method (using VAE and AE, respectively) both mathematically and experimentally. This is very well done and provides the reader with a good understanding of the behaviour of both models. My main concern is the lack of novelty in the proposed method.

I am also slightly concerned that the paper tries to oversell the method a bit. Among the claimed benefits of the method are 1) better density mapping without over-regularised latent variables, 2) single-stage training, 3) minimal reconstruction trade-off, and 4) faster convergence. As far as I understand, benefits 1, 2, and 3 are shared with similar models (VAE+flow) and are as such not unique to GLF. I am not convinced that benefit 4 is true either, as from figure 3, the convergence rates seem to be similar. GLF clearly reaches a better FID, but this seems to happen before epoch 100, which is the earliest shown. It is not clear from the plot if GLF simply starts out being better or when it gains the advantage. Furthermore, in the discussion just below figure 3, the authors note that "even with large $\beta$, GLF still slightly outperforms VAE+flow prior". I find this to be a stretch - had the training stopped at epoch 400, the conclusion would have been that the methods perform identically.

While the authors have clearly put a lot of effort into the paper, they seem to have been rushing for the deadline. There are numerous typos and half-missing sentences (too many to list all, but the worst are pointed out below), so the text needs some polishing before publication. I think it would also make sense to rework section 1 and 2 as, currently, they both present introduction, motivation, and related works.

Questions:
- Just above section 3.2, you say that you add a random permutation after each coupling layer. This is not shown in figure 1(b) if I understand it correctly. Here, only a permutation after the entire block is shown. Did I misunderstand the model?
- Were the model runs in figure 3 also repeated as in table 1? If so, are the standard deviations just too small to be seen or nor shown at all?

Minor comments:
- "Auto-encoder" has inconsistent capitalisation throughout the paper.
- The very first sentence of the introduction misses an ending.
- Fourth sentence of section 2 misses an ending.
- Fifth sentence of section 2 changes the notation - I believe it should be z ~ p(z) here to be consistent.
- There are many examples of a whitespace missing between a reference and the preceding word.
- p 2: "detremine" -> "determine"
- p 4: "assumptio" -> "assumption"
- p 6: "Frchet" -> "Fréchet"



**Experience Assessment:**

I have read many papers in this area.

**Review Assessment: Checking Correctness Of Derivations And Theory:**

I assessed the sensibility of the derivations and theory.

**Review Assessment: Checking Correctness Of Experiments:**

I assessed the sensibility of the experiments.

**Review Assessment: Thoroughness In Paper Reading:**

I read the paper at least twice and used my best judgement in assessing the paper.

---

> ### Author Response · Authors · 2019-11-09
> **On the concerns regarding overselling the model**
>
> We thank the reviewer for the insightful comments. We apologize for the inconvenience caused by our issues in writing, such as typos and missing words. We will carefully check the whole paper and upload a new version soon.
>
> We first address the reviewer's concern regarding the novelty of our work. The reviewer said that our model does not have obvious advantages over VAE+flow prior. However, the point we want to emphasize is that, we are not claiming advantages of our model over VAE+flow prior with LARGE $\beta$. Indeed, to the best of our knowledge, we are the first to observe that VAE+flow prior can generate high quality samples WHEN $\beta$ IS LARGE. As we mentioned in the paper, some previous works discuss VAE+flow prior, but they merely focus on likelihood and they want to show the improvements on test data likelihood over plain VAE, so they do not consider tuning the $\beta$ (i.e., always set $\beta=1$). However, VAE+flow prior with $\beta=1$ does not improve FID scores (see Table 1 in paper). We make the novel finding that generation quality consistently improves as $\beta$ increases. Motivated by this finding, we consider the case when $\beta$ is extremely large, so that the AE is roughly only optimized by reconstruction loss (no randomness in $z$), and the likelihood loss has negligible effects on the AE. We propose GLF based on this idea. We will re-organize Section 3 to make our reasoning more explicit.
>
> In short, we are the first to find VAE+flow prior with large $\beta$ can improve generation quality, and GLF can be seen as a limiting case of this model. Actually, this finding itself is very important (we will make it explicit in the new version), as it can motivate further study on the trade-off between reconstruction and dsitribution matching of VAEs with learnable prior (which, in some sense, similar to beta-VAE paper which can be seen as reducing the reconstruction weight to emphasize distribution matching). Therefore, we do not claim advantages such as faster convergence and better density mapping over VAE+flow prior WITH LARGE $\beta$, as it is also proposed and studied by us. We only claim these advantages of GLF over other methods that aim to generate high quality samples based on AE, as shown in Table 1. However, it is noticeable that GLF does have an important advantage over VAE+flow prior: there is no need to tune $\beta$, which is critical for generation quality, as shown in Figure 3. GLF's slight improvements on FID over VAE+flow prior with $\beta=400$ is just a bonus instead of a highlighted point. We hope this can resolve your concern regarding possible overselling the model.
>
> Answers to minor questions:
>
> 1. Indeed, we only apply a permutation after the entire block. We will correct that in the paper.
>
> 2. In Table 1, the result of GLF on cifar-10 is obtained by training the model for 200 epochs, and the result of VAE+flow prior is obtained by setting $\beta=1$ (as did in previous works) and training for 400 epochs. In Figure 3, we train both GLF and VAE+flow prior with various $\beta$'s for 500 epochs, and we do not include $\beta=1$ because its FIDs are high. We do not include standard deviations in the plot because they are very small in the scale of the plot.
>
> We thank the reviewer again, and we will let you know when the new version is uploaded.

---

> > ### Comment · AnonReviewer2 · 2019-11-15
> > **Thank you for your response**
> >
> > Dear authors, thank you for taking the time to update the paper. It is much improved. And thank you for addressing my questions and concerns.
> >
> > I appreciate that you are putting more emphasis on the observation that the quality of the generated images improves when the reconstruction loss dominates. This is indeed an interesting observation and deserves to be highlighted. However, I still feel the overall novelty is a bit limited. I agree with reviewer 3 that the paper feels closer to engineering.

---

> > > ### Author Response · Authors · 2019-11-15
> > > **Thanks for comments on the revision**
> > >
> > > We thank the reviwer for looking into our response and revised draft, and providing positive feedback on the revision. We do think that our work is a combination of revealing some prrviously ignored phenomenon and proposing a particular method. Therefore, although our model follows naturally from our observations, it is not just an engineering trick. Indeed , the effectiveness of our model further shows the importance of our observations. We hope this work can  also motivate further study on the trade-off within VAEs objective, just as the beta-VAE paper does.

---

### Author Response · Authors · 2019-11-13
**New version uploaded**

We sincerely thank reviewers for their useful comments. We have modified the draft accordingly, and uploaded a new version of the draft. We summarize our revisions here:

1. We rewrite the first three sections to make them easier to follow.  In particular, we revise section 1 to introduce our motivations in a clearer way and emphasize our contributions.  In section 2, we remove some unnecessary words and explicitly divide previous work on modifying AE based models into two categories. Hopefully these will make the section easier to read. We completely re-organize section 3 to make it closely follow our motivation to derive the model and analysis of the model.  In particular, section 3.2 includes our analysis on VAE+flow prior and our finding on the relation between weight of reconstruction loss and sample quality, and section 3.3 clearly derives our GLF model as a limiting case of VAE+flow prior.

2. We present some results of nearest neighbors of generated samples in Appendix G. These, together with the noise interpolation results in the original draft, provide strong evidence that our generative model can generalize to unseen images rather than memorizing the training set.

3. We carefully proofread the whole paper and try our best to fix issues in writing.  We also correct some minor mistakes and add some clarifications, as suggested by reviewers.

Lastly, we make a general comment on our work. Our main contributions are actually two folds: (1) we make the novel observation that the sample quality of VAE+flow prior consistently improves as the weight on reconstruction term increases (2) we propose GLF as a limit of VAE+Flow prior, which is shown to be effective. The first point might be ignored in our previous version due to our presentation, and we modify the draft to emphasize it. We believe this finding itself is interesting, and possibly has some meaningful implications on studying the trade-off between the reconstruction loss and KL loss in VAE's objective.  We hope the contributions can be useful to evaluate the novelty and importance of our work.

We thank the reviewers again, and we thank all readers.

---

### Decision · Program_Chairs · 2019-12-19

**Decision:**

Reject

**Comment:**

The authors propose generative latent flow which uses autoencoder to learn latent representations and normalizing flows to map that distribution. The reviewers feel that there is limited novelty since it is a straightforward combination of existing ideas.